

# A Versatile, Linear Complexity Algorithm for Flow Routing in Topographies with Depressions

Guillaume Cordonnier[1,2], Benoît Bovy[3], and Jean Braun[3]

[1]Univ. Grenoble Alpes, 1251 Avenue Centrale Domaine Universitaire, 38400 Saint-Martin-d'Hères, France
[2]Inria Grenoble Rhône-Alpes, 655 Avenue de l'Europe, 38330 Montbonnot-Saint-Martin, France
[3]GFZ German Research Centre for Geosciences, Telegrafenberg 14473, Potsdam, Germany

*Correspondence to:* Guillaume Cordonnier

**Abstract.** We present a new algorithm for solving the common problem of flow trapped in closed depressions within digital elevation models, as encountered in many applications relying on flow routing. Unlike other approaches (e.g., the so-called "Priority-Flood" depression filling algorithm), this solution is based on the explicit computation of the flow paths both within and across the depressions through the construction of a graph connecting together all adjacent drainage basins. Although this represents many operations, a linear time-complexity can be reached for the whole computation, making it very efficient. Compared to the most optimized solutions proposed so far, we show that this algorithm of flow path enforcement yields the best performance when used in landscape evolution models. Besides its efficiency, our proposed method has also the advantage of letting the user choose among different strategies of flow path enforcement within the depressions (i.e., filling vs. carving). Furthermore, the computed graph of basins is a generic structure that has the potential to be reused for solving other problems as well. This sequential algorithm may be helpful for those who need to, e.g., process digital elevation models of moderate size on single computers or run batches of simulations as part of an inference study.

## 1 Introduction

Finding flow paths on a topographic surface represented as a Digital Elevation Model (DEM) is a very common task that is required by many applications in domains such as hydrology, geomorphometry, soil erosion and landscape evolution modeling, and for which various algorithms have been proposed either for gridded DEMs (e.g., O'Callaghan and Mark, 1984; Jenson and Domingue, 1988; Quinn et al., 1991; Tarboton, 1997) or unstructured meshes (e.g., Jones et al., 1990; Banninger, 2007).

Closed depressions may arise in DEMs because they are real topographic features or result from interpolation error during DEM generation or its lack of resolution. These spurious local minima need to be resolved because they disrupt flow routing, produce hydrologically unrealistic results or introduce artificial singularities that may result from a sudden, unrealistic jump in computing discharge. Flow routing is often corrected by filling depressions (e.g., Jenson and Domingue, 1988), carving channels through artificial sills (Rieger, 1998) or by using hybrid breaching-filling techniques (Lindsay, 2016).

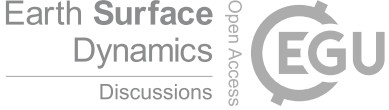

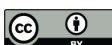

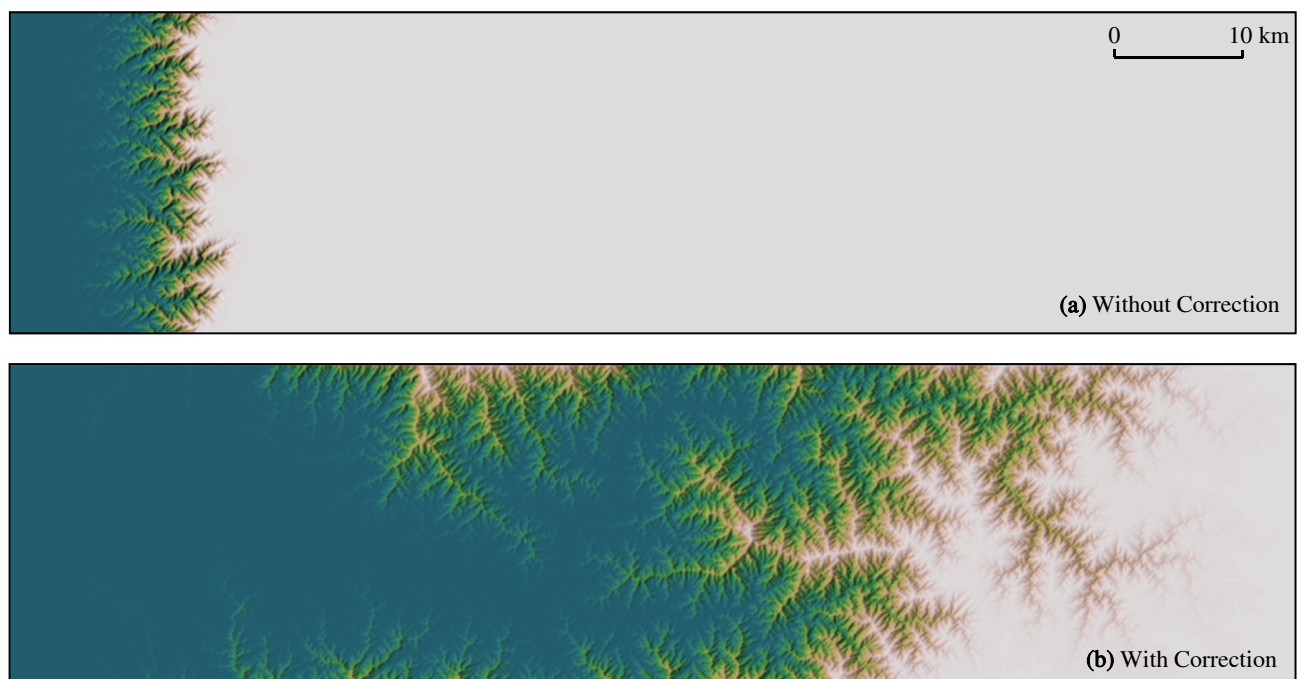

**Figure 1.** Simulation of the evolution of an escarpment over 70 time steps of $1,000$ years each and on a $1024 \times 256$ regular grid, using the FastScape model (Braun and Willett, 2013, see also section 3 below). The grid nodes of the leftmost column have a fixed (base level) elevation while the initial elevation of the other nodes corresponds to a $500$ m high flat surface with small random perturbations. Using the same set of model parameters, simulation results are shown (a) without and (b) with correction of flow routing in local depressions at each time step. As illustrated, flow path disruptions in (a) cause a much slower migration of the escarpment, while the topography predicted in (b) is usually considered as more realistic, especially under temperate or humid climates.

Although not having a linear time complexity, the most recent algorithms of depression removal – e.g., the "Priority-Flood" algorithm and its variants (Barnes et al., 2014a; Zhou et al., 2016; Wei et al., 2018) – have been optimized so that they can deliver acceptable performance when used with large datasets. To increase performance for very large datasets, further optimization efforts have been focused primarily on rather complex, parallel variants of these algorithms (Barnes, 2016; Zhou
5  et al., 2017).

Yet, in some applications flow path enforcement still remains the main bottleneck. This is for example the case in many Landscape Evolution Models (LEMs) simulating an evolving topography (see Tucker and Hancock, 2010, for a review) and that rely on flow routing to compute erosion rates. To produce realistic results, flow path enforcement is often applied many times, i.e., at each simulation time step (Figure 1), even when this eventually becomes irrelevant as the modelled erosional processes
10  usually tend to remove depressions rather than deepen or add new ones (Braun and Willett, 2013). Furthermore, LEMs are also used as forward models in sensitivity analyses and/or inferences on the parameters that control erosional processes, which often





require running a large number of models to adequately explore the parameter space. Parallel flow routing and hydrological correction algorithms don't help much here, as grid-search and/or sampling methods (e.g., Sambridge, 1999) are generally easier to implement and more effective to execute in parallel. Highly optimized, sequential algorithms are still needed in this case.

We have developed a new method of flow enforcement that is based on the explicit building of a graph of drainage basins (possibly encompassing depressions) and the computation of the flow paths both within inner and across those basins. This idea was first introduced in a *Computer Graphics* implementation of the Stream Power Law (Cordonnier et al., 2016), but with a sub-optimal complexity. Although this approach may appear naive at first glance, we have improved it by using of fast algorithms of linear complexity at each step of the procedure, which now makes the whole computation very efficient. Not only
this method allows using a wide range of techniques of flow enforcement within the closed depressions (e.g., depression filling, channel carving or more advanced techniques); but it also provides generic data structures that could potentially be reused for solving other problems like modelling the behavior of erosion/deposition processes within those depressions.

After a detailed presentation of the method and each of its stages, we will show in the sections below through some results how our algorithm behaves and performs compared to existing solutions of flow path enforcement. We will finally discuss the
assets and limitations of our method, with some focus on landscape evolution modelling applications.

## 2 Algorithm

The *input* of the algorithm is a *topography* $\mathcal{T} = (\mathcal{N}, \mathcal{E})$, where $\mathcal{N}$ is a set of nodes and $\mathcal{E}$ is a set of edges that link pairs of neighbor nodes. A node n is given a horizontal position $\boldsymbol{p_n}$ and a vertical elevation $z_n$. A topography may for example result from a triangulation or correspond to a regular grid of 4-connectivity (i.e, four neighbors per nodes) or 8-connectivity
(i.e., also including diagonal neighbors). We follow the conventions of Braun and Willett (2013) to define flow paths on the topography: each node n is given (1) a single flow receiver, $rcv(n)$, which corresponds to the one of its downslope neighbors having the steepest slope, and (2) a set of flow donors, $Donors(n)$, which is a subset of the neighbors of n such that $z_k > z_n$, and $rcv(k) = n, \forall k \in Donors(n)$. We set $rcv(n) = \varnothing$ when n is a *singular node*: it either corresponds to a user-defined *base level* node (e.g., a node on the domain boundary) or a *local minimum* in the topography, i.e., a node inside the domain where
all of its neighbors have a higher elevation, also called a *pit* using Lindsay (2016) terminology.

We propose an algorithm that updates the receivers of a subset of $\mathcal{N}$ such that the flow is never trapped in local minima. This algorithm primarily aims at resolving local minima in the context of flow routing and thus leaves the elevation of the nodes unchanged. Hence it breaks the previously introduced definition of a flow receiver: the new receivers assigned by the algorithm generally produce some localized "upslope flow". While this seems unnatural and may not be wanted, the data structures used by the algorithm provide enough information to efficiently address this issue later depending on the application, which
is beyond the scope of this work. Still, the algorithm ensures that the updated flow routing stays consistent across the whole topography by respecting the following properties:



1. Given a node n, there exists a single base level node b, and a unique flow path from n to b. The flow path is defined as the set of nodes $P = (n, rcv(n), rcv(rcv(n)), ..., b)$.

2. This flow path contains no cycle, i.e., any node appear only once in P

3. The energy $E$, defined by the sum of the elevations of the nodes in the path (eq. 1), is minimal.

$$E = \sum_{i \in P} z_i \tag{1}$$

Our method is essentially based on the computation of a graph connecting adjacent drainage basins. We define a *basin* as the set of all nodes that flow toward the same singular node (Figure 2 (b)). A basin is either a boundary basin or an inner basin depending on whether the singular node is a base level node or a local minimum. To better explain the problem that we want to solve, let's also define a *depression* in a basin B as a subset of nodes $\in$ B that all have an elevation below a given water level, and the *spill* as the node that defines the water level of B (note that the spill does not always belong to B). Intuitively, the basin water levels correspond to the elevation of the surfaces of the lakes that would naturally appear at steady-state after enough precipitations if the topography was perfectly impermeable. The water level of a boundary basin corresponds to the elevation of its base level node so that it contains no depression. In the case of nested depressions, the water level of a basin may be higher than the elevations of all its nodes.

One may break the problem of flow path enforcement down to three, smaller problems: find the spill of each depression, force the flow within the depressions to be routed toward their respective spill and ensure that the flow through the spills is properly routed into adjacent basins. The proposed algorithm addresses this problem in an explicit manner and can be divided into three main stages:

1. Compute the basins and link all pairs of adjacent basins (Figure 2 (b)).

2. Select only some of the basin links computed at the previous stage and orient them such that the flow is routed consistently across adjacent basins, from inner basins toward the boundary basins (Figure 2 (c)). This operation is not trivial: an optimal selection needs a global knowledge of the whole basin graph. To do so, we use an algorithmic structure: a *minimum spanning tree* of the basin graph. We propose here two algorithms, a simple one with $O(n \log n)$ complexity, and a more complex one with $O(n)$ complexity.

3. Update the flow receivers. Using the links selected at the previous stage, we update (only some of) the receivers to enforce the flow both within and across inner basins so that it is ensured to finally reach the boundary basins and their base level nodes. We propose three different methods (one may choose a method over another depending on the specific problem to solve).

Those stages are each detailed in the next sections.




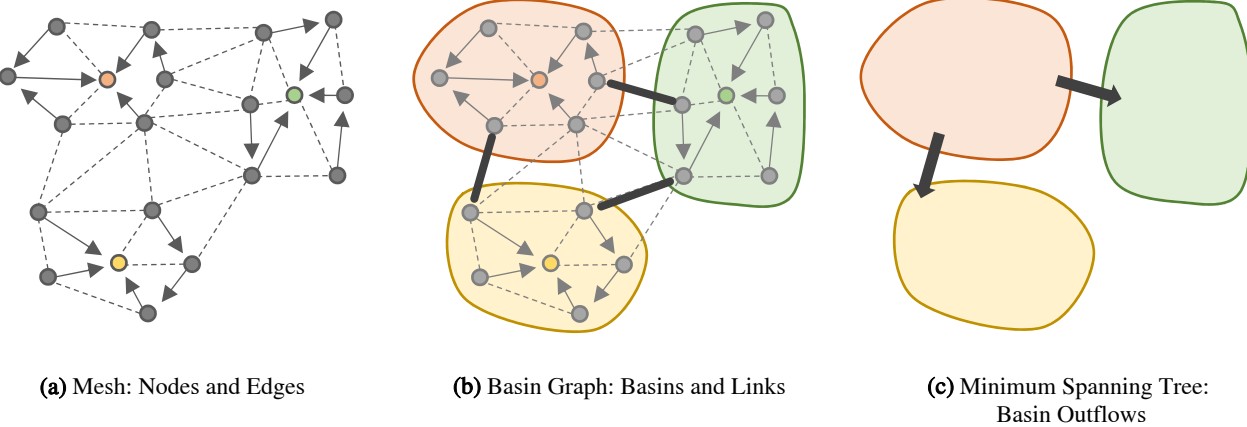

**(a)** Mesh: Nodes and Edges  **(b)** Basin Graph: Basins and Links  **(c)** Minimum Spanning Tree:
Basin Outflows

**Figure 2.** Illustration of the inputs and the first steps of the proposed flow routing algorithm. (a) The input topography is defined on top of a mesh by a set of nodes and edges. A single edge is selected for each node, it connects the node to its flow receiver, i.e., its neighbor with the steepest slope. Nodes with no receiver are local minima (colored in the figure). (b) All the nodes that flow to a same local minimum belong to the same basin. A graph of basins is created by connecting together adjacent basins with *links*, which are materialized on the mesh by edges representing the passes, i.e., the crossings of lowest elevation that connect each pair of basins (black thick arrows). (c) Some of the links are selected by computing a minimum spanning tree and the corresponding passes are oriented in the direction of the flow across the basins (unidirectional arrows). This structure is then used to update the flow receivers so that the flow reaches the domain boundaries without being interrupted.

## 2.1 Basin computation and linkage

This stage consists in first assigning a basin identifier, $\mathrm{basin\_id(n)}$, to each node n of the topography. The identifiers are added sequentially by starting at singular nodes and visiting the donors recursively (see appendix A1).

Then, the *links* connecting all pairs of adjacent basins are retrieved. To each link also corresponds an edge of the topography, here called a pass, which represents the crossing of lowest elevation between the two basins. For example, the link $L = (B_1, B_2)$ connects the basins $B_1$ and $B_2$ and has the corresponding $\mathrm{Pass}(L) = (n_1, n_2)$, where $n_1 \in B_1$ and $n_2 \in B_2$ and where the chosen $(n_1, n_2)$ minimizes $z_{\mathrm{pass(L)}} = \max(z_{n_1}, z_{n_2})$. We define a single procedure to retrieve both the links and their pass (see appendix A2). This procedure parses each edge of the topography: if the two nodes of the current edge have each different basin identifiers, then (1) it adds a new link if no link has been already set for these two basins, and (2) it sets or maybe updates the pass of that link with the current edge.

The sets of basins $\mathcal{B}$ and the set of retrieved links $\mathcal{L}$ both define a *basin graph*. It is worth noting that, at this stage, the links/passes are not oriented and that only one link/pass is stored for two adjacent basins. The procedure described above runs sequentially and won't add the link $(B_2, B_1)$ if it already added the link $(B_1, B_2)$.





## 2.2 Flow routing across adjacent basins

To illustrate the problem, let's start from an inner basin. If it is filled with water, the water level will raise until it finds a pass where water eventually pours into another, adjacent basin. The associated link is then called the *outflow* of the basin. Hence, routing the flow across the basins consists in connecting all outflows such that the resulting flow paths, from inner basins to the

boundary basins, have the same properties than stated above, i.e., those paths are unique, contain no cycle and minimize the energy needed to reach the boundary basins.

If we add to the basin graph a virtual basin (let's call it *external basin*) to which we link all the boundary basins (i.e., the external basin may be viewed as a bucket collecting all the flow that leaves the domain), then we can represent the connected outflows using a specific algorithmic structure: a tree. More specifically, a *basin tree* is a tree that satisfies the properties above;

it actually corresponds to a *minimum spanning tree* of the basin graph, i.e., a subset of the basin graph resulting from a selection of the links so that the following energy is minimized:

$$E_{tree} = \sum_{L \in \mathcal{O}} z_{\mathrm{pass(L)}} \tag{2}$$

Where $\mathcal{O}$ is the set of selected links (or the set of outflows) and $z_{\mathrm{pass(L)}}$ is the elevation of their respective passes.

We propose two algorithms for the computation of a minimum spanning tree. *Kruskal*'s algorithm is very generic and simple

with a log-linear complexity. We also propose a second algorithm, which leverages the planar nature of the basin graph to reach a linear complexity.

---

**Algorithm 1** Kruskal

---

**for** each Basin B  **do**
    MakeSet(B)
**end for**
Sort all links $\mathcal{L}$ by increasing weight $z_{\mathrm{pass(L)}}$
**for** each Link L = $(B_1, B_2)$  **do**
    **if** Find($B_1$) $\neq$ Find($B_2$) **then**
        Add $(B_1, B_2)$ to the Basin Tree
        Union($B_1$, $B_2$)
    **end if**
**end for**

---

### 2.2.1 Kruskal's algorithm

Kriskal's algorithm (Kruskal, 1956) is one of the most classical algorithm used for computing minimum spanning trees and is known to have a $O(m \log m)$ complexity, where $m$ is the number of links. The number of links being always majored by

(c) Author(s) 2018. CC BY 4.0 License.





the number $n$ of nodes in the grid, using this algorithm come a global upper bound of $O(n \log n)$ on the complexity. This algorithm uses a *Union Find* structure to store and merge equivalence classes of objects (see Algorithm 1). The idea here is to parse all links $L \in \mathcal{L}$ sorted by increasing elevation $z_{\text{pass(L)}}$, progressively grouping each pair of basins as a larger, virtual one (equivalence class). All subsequent paths between basins within this equivalence class are discarded to prevent loops. The

Union Find data structure has three operations:

**MakeSet** Create an equivalence class containing a single element.

**Union** Merge two equivalence classes.

**Find** Get the equivalence class of an object.

The optimal implementation of the Union Find structure provides a $\mathcal{O}(\alpha(N))$ complexity for these operations, where $N$

is the number of elements in the structure (i.e., here the number of basins) and $\alpha$ is the inverse Ackermann function whose complexity is lower that $O(\log N)$. This however requires first sorting the links by increasing weight (i.e., by the elevation of their respective passes), which finally yields a $O(m \log m)$ complexity for the whole computation.

### 2.2.2 Planar graphs

The problem of finding the minimum spanning tree is known to have a $O(N)$ complexity when the graph is *planar* (Mareš,

2002). A planar graph is a graph which can be embedded in a plan such that none of its edges cross another one. The basin graph described in section 2.1 is an example of planar graph. The key intuition behind the algorithm proposed in Mareš (2002) is that at least half of the vertices of a planar graph have at most 8 neighbors. The algorithm is then an adaptation of another classical algorithm, named *Boruvka's algorithm* (Boruvka, 1926) ; see Algorithm 2 for more details. The $O(N)$ complexity comes from the fact that at each step of the outer loop, we parse and remove at least half of the nodes of the graph, and

$\sum N + N/2 + \cdots + 1 < 2N$. As the number of grid nodes $n > N$, the complexity of this algorithm is bounded by $O(n)$.

---

**Algorithm 2** Planar Boruvka (Mareš, 2002)

---

Initialize the basin tree structure

**while** There remains nodes in the basin graph **do**

    **while** There is a basin B that has less than 8 neighbors **do**

        Add the link with the lowest neighboring pass to the basin tree

        Contract the link (if the link connects basins B and $B_p$, remove B and append all remaining neighbors of B to $B_p$)

    **end while**

    Clean the graph: bucket sort all links lexicographically to remove parallel edges.

**end while**

---

A special case may arise when the basin graph is computed from a grid of 8-connectivity. In this case, the edges of the graph may cross each other due to the diagonal connectivity, possibly making the basin graph not perfectly planar. This is, however,



rather unlikely as it implies that two passes connecting different basins are found on the two diagonals connecting four adjacent nodes of the grid. In fact, we may still apply this algorithm if we assume that half of the basins in the basin graph are linked to at most 16 adjacent basins – an upper bound that we have validated experimentally – at any step of the algorithm.

## 2.3 Updating flow receivers

The basin tree obtained at the previous stage must be oriented before routing the flow from inner basins to the boundary basins. This is achieved by traversing the tree in the reverse order (i.e., starting from the boundary basins) and labelling the two nodes of each pass, one as $n_{in}$ (incoming flow) and the other one as $n_{out}$ (outgoing flow). Depending on their elevation, either $n_{in}$ or $n_{out}$ is the spill node of the corresponding basin.

The last stage then consists in updating the flow receivers so that any flow entering an inner basin is ensured to leave the
10 basin through $n_{out}$. The most straightforward solution would be to only update the receiver of each local minimum $p$ so that $rcv(p) = n_{out}$. Note that if $n_{in}$ has a higher elevation than $n_{out}$, then two receivers must be updated: $rcv(n_{in}) = n_{out}$ and $rcv(p) = n_{in}$. This very simple solution ensures topological continuity of the flow but does not preserve its spatial continuity. We therefore propose two other, more realistic methods: one similar to *depression filling* and another similar to *depression carving*.

### 2.3.1 Depression carving

The idea is here to mimic the effect of a river carving a narrow trench between the bottom of the depression and the spill: a new, single path is computed from the local minimum to the pass. In fact, the most direct path is already defined by the flow receivers that were computed initially, but it is in the reverse order, i.e., from the pass to the local minimum. Hence, It is trivial to follow this path and progressively revert the receivers until the local minimum is reached (see Algorithm 3).

---

**Algorithm 3** Depression carving

$n_{cur} = n_{in}$
$n_{next} = rcv(n_{in})$
**while** $rcv(n_{cur}) \neq \varnothing$ **do**
    $n_{tmp} = rcv(n_{next})$
    $rcv(n_{next}) = n_{cur}$
    $n_{cur} = n_{next}$
    $n_{next} = n_{tmp}$
**end while**

---

### 2.3.2 Depression filling

Unlike the previous method, we update here the receivers as if the depressions were completely filled by some material. We define a procedure that starts at a pass and then recursively parses all neighbor nodes in a breadth-first order as long as these





are below water level, at the same time updating the receiver of the current parsed node as being one among its neighbors that has already been parsed (see Algorithm 4). We repeat this procedure for all depressions by traversing the basin tree from the boundary basins to the most inner ones so that accurate water level values can be computed during the procedure. Receivers are chosen according to a cost function that we define here as the minimal distance between a node and $n_{out}$. This function

5   does not yield the perfect path patterns that one would obtain using an euclidean distance function with obstacles on a regular grid, but it is simple and efficient while being accurate enough.

---

**Algorithm 4** Depression filling

---

$cost = n \mapsto \|n - n_{out}\|$

$rcv(n_{in}) = n_{out}$

$queue = \{n_{in}\}$

**while** queue not empty **do**

   n = queue.pop_front()

   $min\_cost = \infty$

   **for** all neighbors $n_{nb}$ of n such as $z_n \geq z_{n_{nb}}$ **do**

      **if** $n_{nb}$ is not in queue **then**

         queue.append($n_{nb}$)

      **else if** $n_{nb}$ has already been parsed and $cost(n_{nb}) < min\_cost$ **then**

         $min\_cost = cost(n_{nb})$

         $rcv(n) = n_{nb}$

      **end if**

   **end for**

**end while**

---

## 3   Results

Our algorithm is run under different settings to illustrate its behavior and compare it with some other, state of the art methods. Most of the examples below are shown within the context of landscape evolution modelling, using a simple model of block uplift vs. channel erosion by the Stream Power Law. This model simulates the evolution of the topographic surface $z$, which can be written as follows:

$$\frac{\partial z}{\partial t} = U - K A^m (\nabla z)^n$$

Where $U$ is the uplift rate, $A$ is the drainage area (a surrogate for water discharge), $\nabla z$ the local topographic gradient, and $K$, $m$, $n$ are the parameters of the Stream Power Law. The latter is solved numerically on a 2D regular grid using an implicit

10   scheme of linear complexity (see the FastScape algorithm described in Braun and Willett, 2013). As common settings, we use $K = 7 \times 10^{-4} \, \text{m}^{0.2} \text{y}^{-1}$, $m = 0.4$ and $n = 1$. Grid spacing is 100 m in both directions.



## 3.1 Illustration of the algorithm

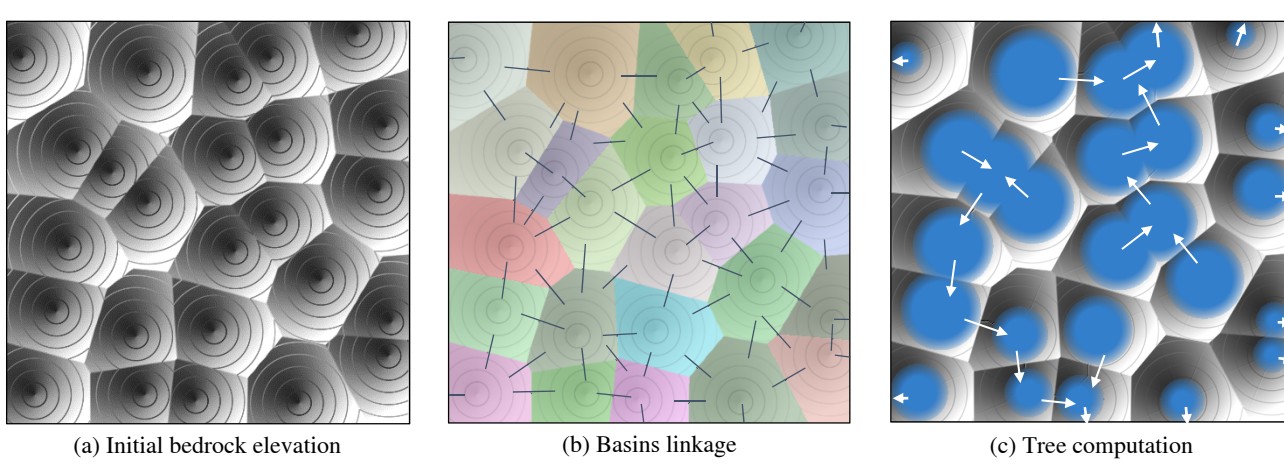

(a) Initial bedrock elevation     (b) Basins linkage     (c) Tree computation

**Figure 3.** Our algorithm of flow enforcement run on a synthetic case. a) Hillshade and contour plot of the input topography with apparent depressions. b) basins (areas of unique, random colors) and all passes connecting adjacent basins (thin black lines). c) Flow directions across the basins (white arrows), as resulting from the computation of a minimum spanning tree from the basin graph, and water level (blue areas) after some erosion is applied on the input topography.

The behavior of our algorithm of flow path enforcement is best illustrated using a simple synthetic topography as input. A set of 25 local minima are sampled on a regular grid of $500 \times 500$ nodes and the elevation of the topographic surface is locally computed as a fixed proportion of the distance to the nearest local minimum (Figure 3 (a)). The first step of the algorithm

delineates the basin of each local minimum and finds all possible connections (links) between the basins, located at the lowest pass between each pair of adjacent basins (Figure 3 (b)). Then, a minimum spanning tree is computed from the graph of these links to find the path of minimum energy that would allow the water to leave the basins (white arrows in Figure 3 (c)). The flow receivers can then be updated by using the edges of this basin tree. The updated receivers are in turn used by the FastScape algorithm to slightly erode the basin boundaries during 1 time step of 100 years. The result is shown as well as the final water

level in Figure 3 (c).

## 3.2 Effect of flow path enforcement strategies on eroded topographies

Figure 1 already shows the effect of flow path enforcement vs. no enforcement on the evolution of an escarpment under active erosion processes. A second set of experiments, shown in Figure 4, illustrates the impact that the different strategies of flow receiver updating have on the evolution of the topographic surface under the action of channel erosion. The input synthetic

topography is defined on a $100 \times 100$ regular grid and looks like an inverted pyramid with $45°$ regular slopes, forming a single, big depression (Figure 4 (a)). The node at the middle of the top boundary is the only node that is not part of the depression: it




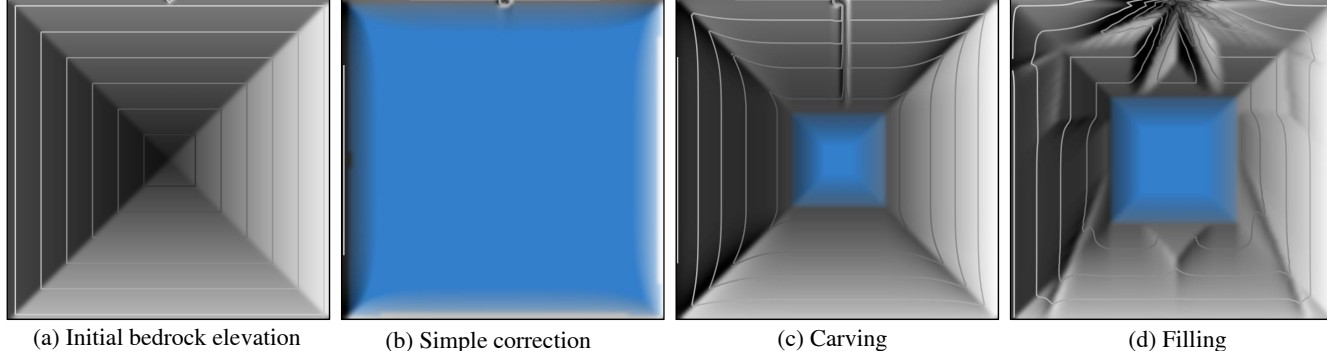

|  (a) Initial bedrock elevation  |  (b) Simple correction  |  (c) Carving  |  (d) Filling  |

**Figure 4.** Demonstration of the effect of flow path enforcement on erosion, using different strategies of flow receivers "correction" within inner basins. a) Hillshade and contour plot of the initial topography. b), c) and d) Hillshade and contour plot of the topography obtained after running a single time step of $5,000$ years with channel erosion only (no uplift), with flow receivers updated using each of the different strategies described in section 2.3. Water level is shown in blue.

has same elevation than the node at the center of the grid and it is defined as a base level node. A single time step of 5000 years of erosion only (no uplift) is performed using each of the strategies described in section 2.3:

**Simple correction.** In this specific case, the algorithm updates the receivers of only three nodes: (1) one of the neighbors of the base level node, which here corresponds to the spill of the closed depression, (2) one of the neighbors of the spill that, together with the spill, forms the pass connecting the depression to the base level node and (3) the local minimum at the bottom of the depression. The new assigned receivers are respectively for (1) the base level node itself, (2) the spill and (3) the other node of the pass. We can see in Figure 4 (b) that this strategy doesn't allow channel erosion to propagate much from the base level node into the closed depression. In fact, drainage area values close to the base level node are high enough to trigger erosion but the low values of drainage area in the vicinity (within the depression) prevents further propagation of the erosion wave.

**Depression carving.** Unlike the former strategy and as expected, Figure 4 (c) shows that the depression carving strategy allows erosion to propagate toward the local minimum along a narrow and deep trench.

**Depression filling.** Using the depression filling strategy, flow receivers are updated over a large area of the depression as if the water surface was replaced by a very gentle slope toward the spill. As a result, erosion affects a great part of the modelled domain, with the emergence of a star-like pattern centered at the spill (Figure 4 (d)). The number and disposition of the branches of the star are due to the grid 8-connectivity used here.

Choosing one strategy over another greatly depends on the specific application. For example, the simple correction strategy may be acceptable if one assumes that no erosion could happen in depressions below the water level. However, interrupted drainage area patterns within the depressions may be problematic when used with erosion algorithms like the FastScape model, which uses an implicit time scheme for solving the Stream Power Law but still treats drainage area explicitly, resulting in too





slow opening of the closed depressions by erosion. The depression carving or depression filling strategies generally yield better results in the latter case. These two strategies have, however, contrasted behaviors and choosing one or the other will depend on several criteria such as the size (i.e., depth vs. volume) of the depressions.

## 3.3 Performances

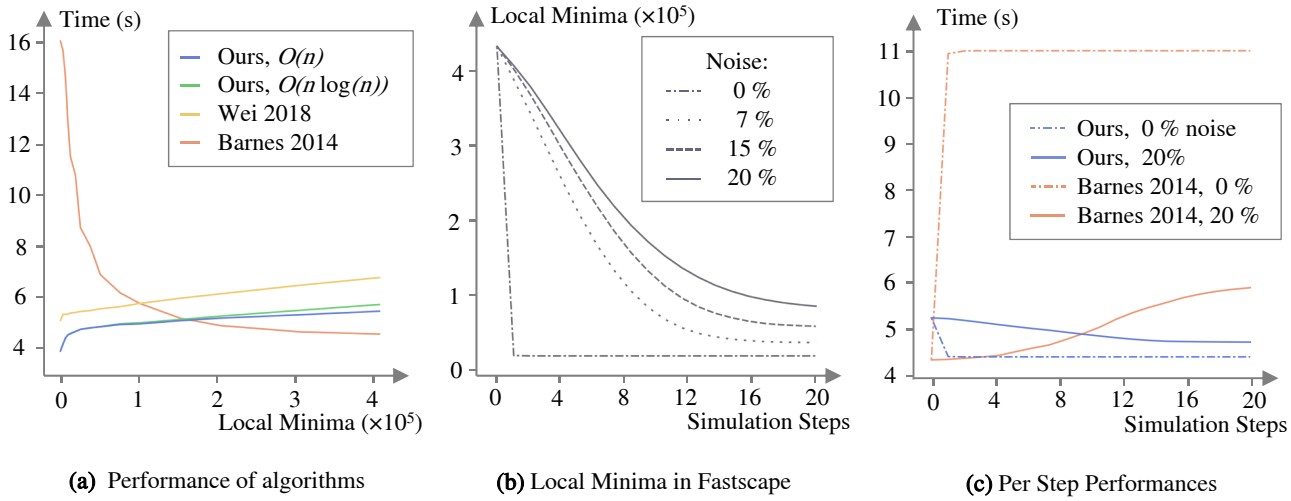

| (a) Performance of algorithms | (b) Local Minima in Fastscape | (c) Per Step Performances |

**Figure 5.** Results from benchmarks assessing the performance of our algorithm for local minima resolution – including both $O(n \log n)$ Kurskal's and $O(n)$ Mareš' variants for computing the minimum spanning tree, compared to two other solutions based on variants of the Priority-Flood depression filling algorithm proposed by Barnes et al. (2014a) and Wei et al. (2018), respectively. See text for more details about the setup of these benchmarks. a) Execution time measured for local minima resolution applied once on a synthetic input topography vs. total number of local minima generated in the input topography. b) Evolution of the number of local minima detected in the topography obtained at each of the first 20 time steps of a FastScape model run. Each curve corresponds to a given magnitude of random perturbations added to produce spatially variable uplift rates (magnitude values are relative to a fixed uplift rate of $5 \times 10^{-3} \text{ m y}^{-1}$). c) Execution time measured for local minima resolution at each time step, with either spatially uniform or variable uplift rates (i.e., a relative noise magnitude of either 0% or 20%). The blue curves refer to our algorithm using the $O(n)$ variant for computing the minimum spanning tree.

5   To assess the performance of our algorithm, we have run multiple benchmarks under various settings. Although these benchmarks mostly take place in the framework of landscape evolution modelling, they provide results that may be useful in other applications too. Note that for better readability, we present here only the results from benchmarks applied on a fixed grid of $4096 \times 4096$ nodes. We obtain consistent results for other grid sizes.

We have run benchmarks for our algorithm – including the two variants for computing the minimum spanning tree but con-
10   sidering only the depression filling strategy – as well as for two other, state-of-the-art algorithms of local minima resolution: one proposed by Wei et al. (2018) and another one proposed by Barnes et al. (2014a). Both of those algorithms fill the depressions using improved variants of the "Priority-Flood" algorithm that reduce the number of nodes processed by a priority queue.





The Barnes et al. (2014a) variant used here, i.e., "Priority-Flood+$\epsilon$", is only slightly optimized but has the advantage of filling the depressions with a nearly-flat surface so that flow directions can be determined. While being the most optimized, sequential variant that has been proposed so far, the Wei et al. (2018) variant fills the depressions with perfectly flat surfaces and thus has to be combined with a flat resolution algorithm – we use here an optimal, $O(n)$ algorithm proposed by Barnes et al. (2014b).

Both variants fill the depressions by directly updating the elevation values on the grid. To ensure proper comparison with our algorithm, we thus need to run them on a temporary copy of the elevation values before computing the flow receiver for each node of the grid. Both the algorithms and the benchmarks are implemented using the C++ language. For the state-of-the-art algorithms, we reuse the implementations available in the RichDEM library v2.2.9 (Barnes, 2018). The benchmarks where computed on an Intel Core i7-4710MQ CPU (2.5 GHz, 32.0 Go RAM). We used Microsoft Visual Studio compiler with fast

optimization options.

In a first set of benchmarks, we create an input topography by running the FastScape model (starting from an initial flat surface with small random perturbations) until steady-state is reached (the uplift rate is set to $5 \times 10^{-3}$ m y$^{-1}$), and then by lowering the elevation of an arbitrary number of nodes down to $10^{-5}$ m below their lowest neighbors. Those nodes are chosen randomly on even rows and columns to make sure that we obtain the same number of local minima in the input topography.

Note that each generated basin have a size of at most 9 cells, which allows for a fine control on the cumulative size of the depressions. Figure 5 (a) shows the execution times of our vs. state-of-the-art algorithms for an increasing number of local minima. We can see that in these settings our algorithm (both variants for the computation of the minimum spanning tree) globally outperforms the state-of-the-art solution of Wei et al. (2018) combined with flat resolution. By contrast, the Barnes et al. (2014a) Priority-Flood variant shows an inverse trend: it performs much worse in the absence of depression but the

execution time rapidly decreases when increasing the total number of local minima, eventually achieving better performance than our algorithm. This is explained by the very simple implementation of this variant, in which all the nodes are processed by a priority queue in the absence of depression (not optimal) while a plain queue is used for most of the nodes if the topography is largely covered by depressions, making this variant near optimal in that specific case. Note that for high numbers of local minima we also start to discern in Figure 5 (a) the difference in performance of the variants used to compute the minimum

spanning tree, here explained by their log-linear vs. linear complexity.

In a second set of benchmarks, we analyze the performances of the algorithms for local minima resolution through full simulations of landscape evolution. We run the FastScape model over 20 time steps of each 10,000 years, starting from a flat topography with small random perturbations (thus containing many local minima) and using fixed boundary conditions, i.e., base level nodes all along the grid boundaries. The simulations are all based on a uniform uplift rate of $5 \times 10^{-3}$ m y$^{-1}$

but each differ by the magnitude of the random field (created on a coarser, $1500 \times 1500$ grid) added to produce spatially variable uplift rates. This magnitude ranges from 0 % to 20 % of the uniform uplift rate. As shown in Figure 5 (b), a greater magnitude of perturbation of uplift rates reduces the rate at which the local minima disappear under the action of channel erosion as the simulation proceed. With no perturbation, all local minima are removed after the first time step. This has important implications for the overall time spent on resolving local minima during a simulation. Figure 5 (c) shows that, with




uniform uplift, our algorithm greatly optimizes this overall time compared to the Priority-Flood variant of Barnes et al. (2014a).
Even with variable uplift rates, our algorithm performs better after only a few time steps.

## 4 Analysis

We focus our discussion on an in-depth analysis of the differences in performance obtained by the different state-of-the-art
algorithms, as reported in the section above. Barnes et al. (2014a) propose to progressively flood the topography from exterior
to interior, keeping in a priority queue all the parsed nodes except for the ones in depressions, which are processed using a plain
queue. The operations used in this algorithm can be split in two main categories: one handling the $n_d$ nodes in depression areas,
with $n_d < n$ the total number of nodes, and another one handling the other "regular" nodes, $n_r = n - n_d$, using the priority
queue. As a depression encloses at least one node (a local minimum) and zero or more nodes in the immediate vicinity, the total
number of nodes in depression areas is always greater or equal to the total number of local minima $n_l$, that is, $n - n_d \leq n - n_l$.
Therefore, the complexity of Barnes et al. (2014a) Priority-Flood variant is bounded by $k_0 + k_1 n + k_2 (n - n_l) \log(n - n_l)$,
where $k_0$, $k_1$ and $k_2$ are constants. Due to the very simple formulation of this algorithm, $k_1$ and $k_2$ are very small. The improved
variants of Priority-Flood proposed by Zhou et al. (2017) and Wei et al. (2018) further reduce the number of nodes that are
processed by the priority queue by carefully selecting spill candidates among the regular nodes. In those variants, the total
number of nodes processed by the priority queue becomes nearly proportional to the number of local minima, inverting the
formulation of the complexity that is here bounded by $k_3 + k_4 n + k_5 n_l \log n_l$. Because those variants are more complex, $k_4$
and $k_5$ have higher values.

   We also derive the complexity of our algorithm taking its stages separately. The first and last stages, i.e., the computation
of the basin graph and the update of flow receivers, are both bounded by $k_6 + k_7 n$, with a relatively high value for the $k_6$ and
$k_7$ constants. The second stage, i.e., the computation of the minimum spanning tree, is either bounded by $k_8 n_l \log n_l$ when
using the Kruskal's algorithm or by $k_9 n_l$ when using the algorithm proposed by Mareš (2002), with $k_8 < k_9$. Both expressions
above are valid considering that $n_l \sim N$ (the number of basins) for $N$ large.

   The complexities of the algorithms that we have derived here are all consistent with the benchmark results shown in Figure 5
(a). The difference between the two minimum spanning tree algorithms is visible only for a large number of local minima,
as predicted by their respective asymptotic complexity, while being unnoticeable for low $n_l$ where the other stages of the
processing prevail. Similar expressions obtained for the complexity of our algorithm vs. the solution based on Wei et al. (2018)
are also well illustrated by sub-parallel curves in the figure. The inverse trend observed for Barnes et al. (2014a) solution is
explained by its complexity, where the term $(n - n_l) \log(n - n_l)$ tends towards zero as $n_l$ increases.

## 5 Conclusions

We have presented here a new algorithm for flow path enforcement in topographies with depressions. We have designed this
algorithm within the framework of landscape evolution modelling and we have demonstrated through benchmarks that, in this



scope, it may greatly improve performance compared to other, state-of-the-art solutions. The potential of this algorithm is, however, not limited to landscape evolution models. On a broader scope, the basin graph and its minimum spanning tree are generic structures that other applications may leverage, possibly through derived quantities such as the water level of each depression. We propose here optimal methods to compute those structures and quantities. Despite the fact that our algorithm

is rather complex and requires some work to be properly implemented, it is designed in a composable way such that it is easy to reuse one or several of its components. Adding new features like alternative strategies of flow path enforcement within the depressions would require only little effort, too.

While being versatile, this new algorithm does not provide an universal solution to the problem of flow routing both within and across closed depressions. Perhaps its main limitation is the assumption of single direction flow, i.e., each node has one

unique flow receiver. Adding full support for multiple direction flow (MDF) without loosing in performance is rather difficult and would require a fair amount of re-design work at each of the three stages of the algorithm:

- Basin computation should take into account divergent flow (basin labels are not unique for grid nodes located on drainage divides).

- It should be theoretically possible to route the outflow from an inner basin into more than one of its adjacent basins (this

is currently not possible using a minimum spanning tree computed from the basin graph).

- Alternative, MDF-compliant methods should be implemented to update the flow receivers within the depressions.

Other algorithms like the Priority-Flood don't have that limitation: they act directly on elevation values and don't prevent us from applying MDF flow routing methods on the modified topography.

Another limitation of this algorithm is its sequential implementation. Further work is needed to adapt it so that it could be run

on modern, multi-core and/or GPU-based architectures. Still, many use cases would benefit from the current implementation. These include processing datasets of moderate size on a single computer or running batches of simulations or analysis pipelines, e.g., in the context of sensitivity analyses or inferences on model parameters.

*Code availability.* An implementation of the algorithm presented here will soon be available (upon publication of this manuscript) in the open-source fastscapelib library (https://github.com/fastscape-lem/fastscapelib), which already provides an implementation in C++ (with

Python bindings) of efficient algorithms for landscape evolution modelling.

*Author contributions.* GC designed the algorithm and implemented it including the examples/benchmarks presented in this manuscript, BB also worked on the implementation. GC and BB worked on the redaction of the manuscript with contributions by JB, all authors contributed to fruitful discussions throughout this study, especially JB who provided many test conditions and use cases.





*Competing interests.* The authors declare that they have no conflict of interests.

*Acknowledgements.* TEXT

## Appendix A: Algorithms

### A1    Basin computation

5  Algorithm 5 finds in which basin belongs each node of the grid by assigning them a label. One unique label is defined (here by an integer) for each basin.

---
**Algorithm 5** Basin computation

---
stack = compute_stack (Braun and Willett, 2013)

label = 0

**for** each node n in stack **do**

    **if** $\mathrm{rcv}(n) = \varnothing$ (i.e. singular node) **then**

        label = label +1

    **end if**

    $\mathrm{basin\_id}(n)$ = label

**end for**

---

### A2    Basin Linkage

Algorithm 6 creates the graph of basins by linking together each pair of adjacent basins. It also finds the passes of lowest elevation between those adjacent basins.



---

**Algorithm 6** Basin linkage

---

Links = $\varnothing$

**for** each node n in stack **do**

    **if** rcv(n) = $\varnothing$ (i.e. singular node) **then**

        $b_{cur}$ = basin_id(n)

    **else if** $b_{cur}$ is not the label of a boundary basin **then**

        **for** each neighbor $n_{nb}$ of n with basin label $b_{nb} \neq b_{cur}$ **do**

            link = $(b_{cur}, b_{nb})$

            $zpass = \max(z_{n}, z_{nb})$

            **if** Links.contains(link) **then**

                **if** elevation of the pass of Links(link) $< zpass$ **then**

                    Update link with pass $(n, n_{nb})$ and its elevation $zpass$

                **end if**

            **else**

                Links.add(link) with pass $(n, n_{nb})$ and its elevation $zpass$

            **end if**

        **end for**

    **end if**

**end for**

---

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
