# Peer review of "A Versatile, Linear Complexity Algorithm for Flow Routing in Topographies with Depressions"

_Earth Surface Dynamics, 2018_

## Author Comment (AC1) · 8 Feb 2019

Dear Referee,

Thanks for your time and appreciation of our paper. We will fix the minor issues that you pointed and improve the discussion regarding the two more important points. Here are some more specific answers:

1) It is hard to compare our sequential version to a parallel one, especially because the results mainly depend on the number of processors. This work is mainly motivated by problems where many different landscape evolution models have to be run at the same

time (for example for parameter estimation), or where distinct DEMs can be processed independently. In these cases, running many instances of a sequential algorithm is always faster than a sequential application of parallel algorithms. We did not explore any parallel implementation of our method, although many parts of the algorithm can be trivially vectorized. Further research need to be performed in finding the bottlenecks and benchmarking a parallel version. We will clarify all this in the paper.

2) We fully agree with the referee that the algorithm implementation should be available in a flexible way so that it could be reused in LEM applications as well as other DEM analysis and modeling applications. Although the current features implemented in the Fastscapelib library are mainly related to landscape evolution modeling, we are developing this library with a broader scope in mind, i.e., the analysis and modeling of topographic data. Features like the algorithm presented in this paper are implemented in the library in a composable way such that it could easily be reused in various contexts. For example, the basin graph is implemented in its own class. We also plan to provide top-level functions to apply our algorithm directly on elevation data. For better interoperability with other libraries, we finally provide wrappers for RichDEM's array objects and for some of its routines. For those reasons and for maintenance reasons, we think it is better to include our algorithm within the fastscapelib library rather than in a stand-alone version.

We will of course be happy to discuss both points if you disagree with our answers.

Best regards, The authors.

---

## Referee Comment (RC2) · Anonymous Referee #1 · 12 Feb 2019

I'm satisfied with the answers provided by the authors in the reply. As such I will be happy to accept the paper pending the resolution of the minor issues that I pointed out in my review.

---

## Referee Comment (RC3) · Anonymous Referee #2 · 17 Feb 2019

The authors present a new depression-filling and breaching algorithm. The work seems technically sound and I look forward to seeing this in publication soon.

The only flaw in the work is that the empirical comparisons versus existing work are weak. I discuss this in greater detail below.

I've recommended the paper for "Major Revision" due to this, but, were it not for this point, I think the rest of the revisions would be minor.

Major Comments ======================= 1. This is a really cool idea and a well-written paper - nice work!

[Figure]

2. There are a number of minor grammatical issues in the paper that need sorting. I've identified most of the ones I found, but others may be lurking.

3. The only major flaw with this paper is that the benchmarks seem ill-constructed. Figure 5a is nice, though I think it would be improved by showing Zhou (2017). (If I recall correctly, RichDEM also has an implementation of this, so it should be easy to do.) I think it would also be improved by considering larger datasets, perhaps up to 10,000x10,000. I think Figure 5c needs some real work. Wei (2018) should be included here, no doubt. Figure 5a shows that it has completely different performance characteristics than Barnes (2014a), so this is critical to contextualizing your work. Surely modifying the algorithm to include surface offsets or flow directions isn't too difficult? Otherwise, perhaps Wei et al. or Barnes et al. have such an implementation? Please find a way to address this.

4. Similarly, as discussed below, the Wei, Barnes, and Zhou algorithms are all adapted to large datasets and, when I've looked at the source, written in a very literate style (minimal bit-hacking). This raises the question of whether versions of these algorithms optimized for use on smaller datasets would perform better in your benchmark. I think you've done a good job of showing that your algorithm ranks with the state of the art, but the comparison isn't strong enough to conclude that the design itself is superior. You may or may not consider this worth addressing.

5. Please make the source code available online. It seems to only be available as a pull request at the moment and this is not suitable for publication. Ideally, the source code and associated tests would be available on Github and archived on Zenodo or Figshare.

Detailed Comments =======================

p. 2, Line 3: "have been optimized so that they can deliver acceptable performance when used with large datasets."

p. 3, Line 8: Check grammar. "using of fast algorithms of linear complexity at each step of the procedure, which now makes the whole computation very efficient."

p. 3, Line 9: Check grammar. "Not only this method allows using" [I feel like I've made an abrupt transition between authors; additional proofreading prior to submission would have been good.]

p. 3, Line 15: "modelling". On p. 1, Line 15 you used "modeling". Please choose a single convention.

p. 3, Line 22: "zk>zn, and rcv(k)=n". Could it be that "zk>zn and not rcv(k)=n" or "not zk>zn and rcv(k)=n"? That is, are both conditions necessary, or does one imply the other? You should clarify this.

p. 3, Line 25: Lindsay (2016) also defines a "flat-bottomed depression". In such a depression, no cell is a proper local minimum. Presumably such cells would be labeled as singular nodes due to the manner in which you construct the rcv array, though you don't make this special case explicit in your paper. You should clarify this.

p. 4, Pt 1: This is the inevitable result of using convergent/unidirectional flow routing.

p. 4, Pt 2: Check grammar.

p. 4, Pt 3: I have difficulty following this point. A minimum can only be defined with respect to a set of possible alternatives and it is not clear at this point what the set of alternatives is. Since you haven't introduced it previously, I would appreciate a forward reference to the point in your paper where you do introduce it.

p. 4, Line 8: "whether the singular node is a base level node". I feel like the term "base level node" is unintuitive, since many nodes could be at, or below, a given base level, but be interior to a landscape. The term "boundary node" seems more intuitive.

p. 4, Line 9: "that all have an elevation below a given water level". I find this definition counterintuitive. A basin includes regions outside of a depression (up to the peaks of

mountains). We are now calling a depression the set of cells below a chosen water-level, but nothing about this paragraph constrains a water level to be contained with a Lindsay (2016) typology-style depression. It seems possible to set the spill as a mountain peak and declare the entire DEM a depression. I wonder if there's a more intuitive term that can be used here? Or perhaps a constraint can be added?

p. 4, Line 12: "precipitations" -> "precipitation"

p. 4, Line 15: "respective spill and ensure" -> "respective spill, and ensure". I think the use of an Oxford comma here is useful in alerting the reader that the third item has been reached.

p. 4, Lines 19-30: I think these steps are performed iteratively, but don't see mention of this. Could you please clarify?

p. 5, Line 3: "visiting the donors recursively". I followed the reference to the Appendix, but did not find it to be enlightening since it requires knowledge of the stack construction from Braun and Willet (2013). I think you are doing a depth-first traversal of all of the cells in a basin (those cells whose flow ultimately terminates at a given singular node). If so, stating this directly would make things clearer. The word recursion also implies a particular computational motif—functions calling themselves—which isn't implied by the Appendix. If you are using recursion, the Appendix should be modified. However, this is a risky strategy for larger datasets since some operating systems impose relatively low limits on the depth of recursion.

Algorithm 6: It might be useful to note somewhere that the link is undirected such that Links.contains((n1,n2))==Links.contains((n2,n1)). The current implementation seems to rely on the ordering of the stack to ensure this is true, but in another context it may be useful to relax that ordering.

Algorithm 6: 'if elevation of the pass of Links(link) < zpass then' finds the highest link between two basins. I think this is the opposite of what you want.

p. 6, Line 2: "To illustrate the problem" Which problem?

p. 6, Line 3: "Hence, routing the flow across the basins consists in connecting all outflows such that the resulting flow paths, from inner basins to the boundary basins [...]" As I describe elsewhere in this review, this isn't strictly true and represents a serious restriction on the use of this algorithm. (TODO)

p. 6, Line 18: "Kriskal's algorithm" -> "Kruskal's algorithm" // "classical algorithm" -> "classical algorithms"

p. 6, Line 19: "majored by". I'm not familiar with this terminology.

p. 7, Line 1: "using this algorithm come a global upper bound". Grammar.

p. 7, First paragraph: "Union Find" is usually written as "Union-Find".

p. 7, Line 11: "lower that O(log N)" -> "lower than"

p. 7, Line 15: "in a plan" -> "in a plane"

Algorithm 2: This is super cool! What a good find. The pseudocode here seems as though it won't get you to O(N) time since each step must be implemented carefully to make that guarantee. Perhaps some statement to this effect with a reference to Mareš (2002) is in order?

p. 8, Line 1: "rather unlikely". This approach to correctness seems a bit cavalier. I expect that a good algorithm works in all cases, not merely when it's convenient. The rest of this paragraph seems to imply that, in theory, this problem can be handled. But I'm left with the feeling that your implementation may contain subtle bugs that you've thought are too unlikely to pay attention to! I would like assurance that you check for such exceptional situations and either handle them gracefully and correctly, or flag them for user intervention. It's also unclear why the assumption you state solves the problem and why it isn't possible to build adversarially nested structures.

p. 8, 2.3.1: Are all of the depressions carved to make a continuous flow network? If

so, the situation shown in Figure 3 doesn't seem as though it can happen: Flow will progress along the carved network before water ever pools. Similarly, if depression-filling is applied, pooling won't occur.

p. 9, Line 4: "minimal distance" The minimal euclidean distance ignoring obstacles? I think this could be clearer.

p. 9: Please number equations.

Algorithms 3 and 4: On Page 3, a Donors array was listed as one of the data structures used. Presumably that needs to be updated as well, though it doesn't appear here. Some explanation of this is needed.

Algorithm 4: I struggled somewhat to follow the logic here. Why is a cost function used instead of a simple breadth-first ordering? Perhaps a figure showing what the different flow paths look like would be helpful. The algorithm also seems as though it will terminate at the depression's pit cell. However, the flow pattern to the outlet will then have half of the depression exhibiting convergent flow to the pit cell and half of the depression showing convergent flow to the outlet. This flow pattern is unrealistic if the entire depression is treated as having been filled with flow superimposed on the resultant flat surface.

Figure 3c: The depiction of standing water shown here is at odds with my understanding of what the flow network looks like following carving or depression-filling. Or perhaps erosion fills in the channels connecting depressions, resulting in standing water? But then one wonders what the point of all of this was?

Figure 4: Since the updates presented in the paper so far only affect the receiver network, it wasn't immediately clear to me how the topography had been generated.

p. 11: If only the receiver array is modified, how is the Equation From Page 9 implemented when water flows uphill? Campforts and Grovers (2015, "Keeping the Edge") show that the Stream Power Law as it's used in Fastscape doesn't preserve knickpoints or other topography discontinuities well. The channels carved here seem as though they are representative of the sorts of terrain this critique applies to. Some discussion of modeling philosophy might be useful.

Figure 5: What I find most interesting about this figure is that Wei 2018 is competitive with your results in (a), but gets dropped in (c). Why, in (c), do you choose to compare your work against a slower algorithm?

Figure 5: Could you put (a), (b), and (c) in bold in the caption to help readers find the relevant text?

p. 12 and 13: Given this description, I don't see how your benchmarks will be meaningful. Using the Wei (2018) algorithm in conjunction with the Barnes (2014b) algorithm for the benchmarks of Figure 5a makes a certain sense, but why not simply modify the Wei (2018) algorithm using the PF+e strategy of Barnes (2014a)? Wouldn't this comparison be more fair?

Why does Figure 5c uses Barnes (2014a)? I can't find an explanation anywhere. It seems like comparing against Barnes (2014a) is a poor choice. Barnes (2014a) suggests a path for accelerating Priority-Flood by reducing the number of cells passed through the Priority-Queue. Barnes (2014a), Zhou (2016), and Wei (2018) all present progressively better algorithms for doing this. It's better to think of all of these algorithms as being $O(N + M \log M)$, where M is the number of cells which pass through the priority-queue. M decreases dramatically between Barnes (2014a) and Wei (2018), as shown in Wei (2018, Figure 8). By the time Wei (2018), the $O(N)$ term might even dominate the $O(M \log M)$ term, meaning that Wei (2018) is a linear-time algorithm! Indeed, your Figure 5a suggests this. In any case, Figure 5c seems to overstate your results since it's not a proper performance comparison.

p. 13, Line 8: "we reuse the implementations available in the RichDEM" This seems reasonable, but it should be done with caveats. As you note on p. 2, this set of algorithms has "been optimized so that they can deliver acceptable performance when

used with large datasets." It seems as though your algorithm uses significantly more space than the B, Z, and W algorithms you cite in your paper. The Barnes (2014a) algorithms work in-place using only the priority-queue and a boolean array as extra memory; that is, it requires about O(sqrt(N)) additional space. In comparison, your code has a Receivers array, Donors array (which takes O(8N) space), and Union-Find array, leading to a O(10N) space requirement. Trading space for time is a well-known optimization technique. This raises the question of how the B,Z,W algorithms would perform had they been optimized for small datasets. It also raises the question of how your algorithm would perform on larger datasets. The average size of the test dataset used by Wei (2018), for instance, was 2*10^8 cells - two orders of magnitude larger than the 1500x1500 grid you use and three orders of magnitude larger than the Figure 5a benchmark. I don't think additional tests are necessary, but it should be clarified somewhere that the algorithms you're comparing against are being used for dataset sizes they weren't designed for.

p. 14, Lines 10-15: This is a really nice analysis. However, I'm not sure I agree with it entirely. This line: "Because those variants are more complex, k4 and k5 have higher values." Why is the cost of using a priority-queue higher for the Zhou and Wei algorithms? I would think that k2=k5. I agree that k4>k1. However, one of the reasons priority queues are expensive is because they have poor cache coherence. (Search online for "Latency numbers every computer programmer should know". See also, Luengo-Hendriks (2010)'s discussion of priority queue performance variation.) This means I expect k1<k2 and k4<k5. Wei trades computation (which is fast) for memory management (which is slow); the question then is whether nl is small enough to offset the higher cost of k4.

p. 14, Lines 10-15: Some analysis of the space used by the various algorithms would improve this discussion.

p. 15, Line 17: The Barnes (2014a) algorithm also includes a variant (+FlowDirs) that could be used to construct a receivers array similar to yours. Presumably Zhou and

Wei could be modified in this way as well.

p. 15, Line 20: Barnes (2019, "Accelerating a fluvial incision") maps Fastscape to multi-core architectures and GPUs, showing a 43x speed-up versus serial performance, but also fails to find a depression-filling/breaching algorithm for that environment.

p. 15, Line 24: Please add a link to a Zenodo archive or Figshare copy of your code. This has the advantage of giving it a doi number and assuring readers can find the version used in your publication.

Bibliography: doi numbering is inconsistent. Please include doi numbers whenever possible.

Flat bottomed-depressions: It would be helpful to make a note of the special cases where many cells are singular and physically adjacent. I think your algorithm handles this gracefully, but it would be nice to make that explicit.

For basins with large collection zones and small depressions, overflow might need to go upstream as well as downstream with respect to what's specified in the receiver array. This limits the utility of this algorithm in situations where dynamic flow is being modeled.

---

## Author Response (AR1)

**Cover Letter**

Dear Editor,

We first would like to thank again the two reviewers for their reviews, that greatly improves the quality of the paper. Please find in the following details of our answers (A) to the reviews (Q), and how we addressed them in the manuscript (R). The marked-up manuscript is attached to the end of this document.

Q: Compare with Parallel implementation of Barnes2016 & Zhou2017:

A: It is hard to compare our sequential version to a parallel one, especially because the results mainly depend on the number of processors. This work is mainly motivated by problems where many different landscape evolution models have to be run at the time (for example for parameter estimation), or where distinct DEMs can be processed independently. In these cases, running many instances of a sequential algorithm is always faster than a sequential application of parallel algorithms. We did not explore any parallel implementation of our method, although many parts of the algorithm can be trivially vectorized. Further research need to be performed in finding the bottlenecks and benchmarking a parallel version. We will clarify all this in the paper.

R: Added (Section 3.3) : "Our algorithm being optimized for a sequential usage, we chose, however, not to compare it to the parallel versions of Barnes2016 and Zhou2017."

Q: Code Availability

A: We share the same concerns than the referee about code availability. Although a pull request on GitHub already refers to a git branch that is published online on this platform, we agree that making the code available in a separate repository may be better in order to reproduce the results presented in this paper. However, we are concerned by having to maintain duplicate codes in different repositories. Maintaining individual repositories for every algorithm that we want to include in the fastscapelib library is not a good solution either. This might actually have a negative impact on reproducibility and readability. Therefore, we propose to create a repository specifically for this paper; once the paper is accepted we will archive it (i.e., making it read-only) and redirect users to the fastscapelib library where further development and maintenance will happen.

Q: Page 1, line 9: other problems as well - better specifying 1 or 2 of them.

**R**: Added (Abstract) "such as the simulation of erosion"

Q: Page 3, line 13: "and each of its stages" I will rewrite this part of the sentence

R: Replaced by (Section 1): "After a detailed presentation of the different steps of the method"

—— Review RC3 ———

**Major issues:**

Q: The only major flaw with this paper is that the benchmarks seem ill-constructed. Figure 5a is nice, though I think it would be improved by showing Zhou (2017). (If I recall correctly, RichDEM also has an implementation of this, so it should be easy to do.) I think it would also be improved by considering larger datasets, perhaps up to 10,000x10,000. I think Figure 5c needs some real work. Wei (2018) should be included here, no doubt. Figure 5a shows that it has completely different performance characteristics than Barnes (2014a), so this is critical to contextualizing your work. Surely modifying the algorithm to include surface offsets or flow directions isn't too difficult? Otherwise, perhaps Wei et al. or Barnes et al. have such an implementation? Please find a way to address this.

A: Maybe the referee means Zhou (2016)? Zhou (2017) is about a parallel version of the algorithm, and it would be irrelevant to add it in the benchmarks. On the other hand, Zhou (2016) could definitively be added, although all our benchmarks agree that Wei (2018) performs better, although with a similar pattern. We already have results of benchmarks for both Zhou (2016) and larger datasets (up to 16k x 16k), so it would not be difficult to add them in the paper if needed. Obviously, as pointed by the referee, all these implementations do not include flow direction. Modifying the code to include surface offset or flow direction would represent additional work (design/implementation/testing) that could be published separately, whereas in this paper we preferred to compare our method with state-of-the-art algorithms that have been already published and for which one can find an existing implementation. The suggested modification is, in our opinion, not yet state-of-the-art.

R: Zhou 2016 has been added to Fig 5a; Wei 2018 to Fig 5c. More details on benchmark follows.

Q: Similarly, as discussed below, the Wei, Barnes, and Zhou algorithms are all adapted to large datasets and, when I've looked at the source, written in a very literate style (minimal bit-hacking). This raises the question of whether versions of these algorithms optimized for use on smaller datasets would perform better in your benchmark. I think you've done a good job of showing that your algorithm ranks with the state of the art, but the comparison isn't strong enough to conclude that the design itself is superior. You may or may not consider this worth addressing.

A: We will elaborate on the details below. As stated before, we now have results for larger grids so we can easily propose a fairer comparison. It should be noted that our goal is more to explore theoretical results (demonstrating the time linearity of our algorithm) than to propose a competitive, optimal implementation.

R: Fig 5 now shows our results on grid of 16k x 16k nodes.

**Detailed comments:**

P. 2, Line 3: "have been optimized so that they can deliver acceptable performance when used with large datasets."

R: Replaced by (Section 1): "have been optimized so that they can be used efficiently on large datasets"

Q: p. 3, Line 22: "zk>zn, and rcv(k)=n". Could it be that "zk>zn and not rcv(k)=n" or "not zk>zn and rcv(k)=n"? That is, are both conditions necessary, or does one imply the other? You should clarify this.

A: Both conditions are necessary: a node is always the receiver of all his donors, and is always strictly below them.

R: (Section 2) We removed the last condition (zk>zn) because it is already included in rcv(k)=n and we changed to a "set" notation: "a set of flow donors, Donors(n), which is a subset of the neighbors of n and is defined as  $Donors(n) = \{k \in Nb(n), \text{ s.t. } rcv(k) = n\}$ "

Q: p. 3, Line 25: Lindsay (2016) also defines a "flat-bottomed depression". In such a depression, no cell is a proper local minimum. Presumably such cells would be labeled as singular nodes due to the manner in which you construct the rcv array, though you don't make this special case explicit in your paper. You should clarify this. A: True, "flat bottomed depressions" are singular nodes in

our terminology. This will be clarified.

R: (Section 2) Changed from "also called a *pit* using Lindsay (2016) terminology." To "and either correspond to a *pit* or a *flat-bottomed depression* in Lindsay (2016) terminology."

Q: p. 4, Pt 1: This is the inevitable result of using convergent/unidirectional flow routing.

A: This is true for unicity, not for existence. The path ends at a base level node (or boundary node), which is in general not the case due to the existence of local minima.

**R**: Added (Section 2): "(not a local minimum)" in point 1.

Q: p. 4, Pt 3: I have difficulty following this point. A minimum can only be defined with respect to a set of possible alternatives and it is not clear at this point what the set of alternatives is. Since you haven't introduced it previously, I would appreciate a forward reference to the point in your paper where you do introduce it.

A: The minimum is chosen among all possible flow paths starting at a given point. This will be rephrased. Another issue with this equation is that this does not hold for nodes below water lever. We will rewrite this condition to properly take this point into account.

R: (Section 2) We prepended for the different properties that they are respected "for each node n of the topography", and changed property 3 to: "The receivers defining P are chosen such that it satisfies properties 1 and 2, and minimizes the energy E, defined as:". We also changed the per node energy to " $E_i$ ", with the following remark: "As a first approximation, we set  $E_i = z_i$  (the altitude of the node). We will discuss later the special case of nodes under water level.". Which is done below, after defining a water level: "The energy of the nodes should be changed to  $E_i = w_i$ , but as described later, one may choose various routing strategies inside the depressions depending on the application. Therefore, we allow any path within depressions by setting  $E_i$  to zero inside them, and keeping  $E_i = z_i$  elsewhere."

Q: p. 4, Line 8: "whether the singular node is a base level node". I feel like the term "base level node" is unintuitive, since many nodes could be at, or below, a given base level, but be interior to a landscape. The term "boundary node" seems more intuitive.

A: We have mixed feelings with this point: "boundary" is clearly more intuitive in the sense of boundary condition, but not in the geometrical meaning. Any boundary condition could easily be chosen inside the domain. Nevertheless, we will do the change.

R: We eventually replaced "base level nodes" by "boundary nodes"

Q: p. 4, Line 9: "that all have an elevation below a given water level". I find this definition counterintuitive. A basin includes regions outside of a depression (up to the peaks of mountains). We are now calling a depression the set of cells below a chosen waterlevel, but nothing about this paragraph constrains a water level to be contained with a Lindsay (2016) typology-style depression. It seems possible to set the spill as a mountain peak and declare the entire DEM a depression. I wonder if there's a more intuitive term that can be used here? Or perhaps a constraint can be added?

A: Linking this definition with Lindsay terminology will certainly make it clearer.

R: (Section 2) We complexly rewrote this paragraph, with more formal definitions: "To better explain the problem that we want to solve, we consider a filled topography as the result of an ideal physical process where a perfectly fluid material has been poured onto an impermeable ground and stabilized at steady state. For a node n, we define as water level  $(w_n)$  the elevation of the fluid surface, and as a *spill* any node s such that  $\exists d \in \text{Donors}(s) | w_d = w_s$  and  $z_s > w_{rcv(s)}$ . Note that for a flow routing observing the aforementioned properties, the water level can be computed as  $w_n = \max(w_{rcv(n)}, z_n)$ . We also use the term *depression* from Lindsay (2016) terminology, and we define it with respect to a basin B as a subset of nodes of B under water level, characterized by  $w_n = w_{rcv(n)}$ . Note that the water level of a boundary basin corresponds to the elevation of its associated boundary node so that it contains no depression. In the case of nested depressions, the water level of a basin may be higher than the elevations of all its nodes, which means that the spill does not always belong to B."

Q: p. 4, Lines 19-30: I think these steps are performed iteratively, but don't see mention of this. Could you please clarify?

A: No, these steps are needed only once to find and resolve all local minima. But each of them include inner iterations over the domain. We will clarify this.

R: (End of Section 2) "Each of these stage processes the whole DEM, and as such are run only once for a given topography."

Q: p. 5, Line 3: "visiting the donors recursively". I followed the reference to the Appendix, but did not find it to be enlightening since it requires knowledge of the stack construction from Braun and Willet (2013). I think you are doing a depth-first traversal of all of the cells in a basin (those cells whose flow ultimately terminates at a given singular node). If so, stating this directly would make things clearer. The word recursion also implies a particular computational motif—functions calling themselves—which isn't implied by the Appendix. If you are using recursion, the Appendix should be modified. However, this is a risky strategy for larger datasets since some operating systems impose relatively low limits on the depth of recursion.

A: We can copy Braun and Willet (2013) algorithm in the appendix if needed (which is indeed a depth first traversal). We will change recursion to traversal which is technically more correct.

R: We changed recursion to traversal wherever it appeared and added the stack computation of Braun and Willet (2013) in Appendix 1

Q: Flat bottomed-depressions: It would be helpful to make a note of the special cases where many cells are singular and physically adjacent. I think your algorithm handles this gracefully, but it would be nice to make that explicit.

R: Added (Section 2.1): "The case of flat bottomed depressions does not require any particular treatment: all nodes within flat areas are singular nodes and therefore are each assigned a unique basin identifier."

Q: p. 6, Line 2: "To illustrate the problem" Which problem?

R: Rephrased to (Section 2.2): "This second stage tackles the problem of selecting the right subset of links so that we obtain consistent flow paths on the basin graph. To illustrate the proposed solution, let's start from an inner basin."

Q: p. 6, Line 19: "majored by". I'm not familiar with this terminology.

R: Changed to "The number of links being always bounded by a linear function of the number n of nodes in the grid (Euler formula)".

Q: p. 7, Line 1: "using this algorithm come a global upper bound".

R: Changed to (Section 2.2.1): "using this algorithm induces a global upper bound of  $O(n \log n)$  on the complexity of our solution.

Q: Algorithm 2: This is super cool! What a good find. The pseudocode here seems as though it won't get you to O(N) time since each step must be implemented carefully to make that guarantee. Perhaps some statement to this effect with a reference to Mareš (2002) is in order?

R: Added (Section 2.2.2): "As demonstrated by Mares (2002), the limit of 8 neighbors for the selection of a basin in the inner loop is critical in halving the number of edges at each iteration of the outer loop and thus in obtaining a linear time complexity."

Q: p. 8, Line 1: "rather unlikely". This approach to correctness seems a bit cavalier. I expect that a good algorithm works in all cases, not merely when it's convenient. The rest of this paragraph seems to imply that, in theory, this problem can be handled. But I'm left with the feeling that your implementation may contain subtle bugs that you've thought are too unlikely to pay attention to! I would like assurance that you check for such exceptional situations and either handle them gracefully and correctly, or flag them for user intervention. It's also unclear why the assumption you state solves the problem and why it isn't possible to build adversarially nested structures.

A: There are several aspects here. First, in 8-connected meshes, the graph may no longer be planar but the crossing edges are in general in low proportion in the overall graph, hence the "rather unlikely" which, indeed, needs more rigorous explanations. By changing 8 to 16 in Mares (2002) algorithm, we validated experimentally that the mathematical condition for the O(n) complexity holds for any (tested) 8-connected graphs. We unfortunately do not have a formal proof for that, but this has no impact on the correctness of the algorithm, only on the complexity. This means that 1) our algorithm is proven to be O(n) for any planar input mesh (e.g. 4 connected or triangulation), and 2) it is not proven that there is no very small chance that for some very rare cases, the complexity might be  $O(n \log n)$  for 8-connected graph. Once again, we never saw this case occurring.

R: Changed to (Section 2.2.2): "Furthermore, this issue does not impact the correctness of the algorithm. Only the linear complexity is not formally proven. Because it is not planar, the case of a 8-connectivity grid would fall in the second category mentioned by Mares (2002) of graphs closed on graph minor. We have validated this experimentally by randomly computing minors of different sized 8-connected graph. We found an edge density of 4, implying that half of the basins in the basin graph are linked to at most 16 adjacent basins (and not 8 as for planar graphs) at any step of the algorithm. Therefore, we have demonstrated the linear complexity for 8-connected graphs experimentally, although future work is needed to prove this in a formal framework."

Q: p. 8, 2.3.1: Are all of the depressions carved to make a continuous flow network? If so, the situation shown in Figure 3 doesn't seem as though it can happen: Flow will progress along the carved network before water ever pools. Similarly, if depression filling is applied, pooling won't occur.

And: Figure 3c: The depiction of standing water shown here is at odds with my understanding of what the flow network looks like following carving or depression-filling. Or perhaps erosion fills in the channels connecting depressions, resulting in standing water?

A: Depression carving and filling are metaphors: our algorithm updates the flow directions, but does not change the elevation of the DEM. We name our variants "filling" and "carving" to match with previous work, but in our case, we compute the flow directions that would appear if the topography has been filled or carved. Pooling can be computed quite easily as a post process on the filling strategy by parsing the flow network from top to bottom (which can obviously not work with carving), and we show it to give an idea of the water level, and hence of the location of the spills on fig 3, and latter to show the effect of the different strategies on erosion in fig 4.

**R**: Added, before 2.3.1: "Note that we use *carving* and *filling* as metaphors as our algorithm only changes the flow graph connectivity without altering elevation values." And, in caption of fig 4.: "Water level is shown in blue, and is computed by propagating the spill elevation while parsing the nodes in the upstream order (based on updated donors)."

Q: Algorithms 3 and 4: On Page 3, a Donors array was listed as one of the data structures used. Presumably that needs to be updated as well, though it doesn't appear here. Some explanation of this is needed.

R: Added, before 2.3.1: "For each of the variants, the donors and stack structures need to be updated to reflect the changes in the receivers."

Q: p. 9, Line 4: "minimal distance" The minimal euclidean distance ignoring obstacles? I think this could be clearer.

R: Added (Section 2.3.2): "Euclidean".

Q: Algorithm 4: I struggled somewhat to follow the logic here. Why is a cost function used instead of a simple breadth-first ordering? Perhaps a figure showing what the different flow paths look like would be helpful. The algorithm also seems as though it will terminate at the depression's pit cell. However, the flow pattern to the outlet will then have half of the depression exhibiting convergent flow to the pit cell and half of the depression showing convergent flow to the outlet. This flow pattern is unrealistic if the entire depression is treated as having been filled with flow superimposed on the resultant flat surface.

A: There are several issues with a filling led by a breadth first traversal, the first one being that it depends on the order of that parse. Local methods generally suffer from straight patterns resulting from the 4 or 8 connectivity, which is why we propose a better approximation with a cost function associated to a simple Euclidean distance. This is not a major contribution of our work, and any other solution could be explored here. There is indeed a mistake in the notation. It should be: "for all neighbors  $n_{nb}$  of n such as  $z_{n_{out}} \ge z_{n_{nb}}$ "

R: Added (Section 2.3.2): "We prefer this method over a simple breadth-first search, which depends on the order in which neighbors are visited and which leads to more pronounced straight lines after erosion, due to the 4 or 8 connectivity.". We also corrected the algorithm.

Q: p. 11: If only the receiver array is modified, how is the Equation From Page 9 implemented when water flows uphill? Campforts and Grovers (2015, "Keeping the Edge") show that the Stream Power Law as it's used in Fastscape doesn't preserve points or other topography discontinuities well. The channels carved here seem as though they are representative of the sorts of terrain this critique applies to. Some discussion of modeling philosophy might be useful.

A: We will add a reference to Campforts and Grovers (2015) to discuss on the limitation of this strategy. We simply follow the fastscape implementation, clamping the slope to 0 to avoid uphill flow. Lake deposition or any other processes could be applied in this case but are outside the scope of this work.

R: Added (After Eq 3): "In particular, the local gradient  $\nabla z$  is chosen as the slope between the eroded node and its receiver (forced to 0 if its value is negative in order to avoid erosion artifacts in the case of "upslope flow" caused by updated receivers). We choose this algorithm which is particularly well suited to our flow routing method, although some discussion on the limits of this algorithm can be found in (Campforts and Grovers, 2015) for steep topography."

Q: Figure 5: Could you put (a), (b), and (c) in bold in the caption to help readers find the relevant text?

**R**: Numberings have been put in bold in all captions.

Q: Figure 5: What I find most interesting about this figure is that Wei 2018 is competitive with your results in (a), but gets dropped in (c). Why, in (c), do you choose to compare your work against a slower algorithm?

And: Why does Figure 5c uses Barnes (2014a)? I can't find an explanation anywhere. It seems like comparing against Barnes (2014a) is a poor choice. Barnes (2014a) suggests a path for accelerating Priority-Flood by reducing the number of cells passed through the Priority-Queue. Barnes (2014a), Zhou (2016), and Wei (2018) all present progressively better algorithms for doing this. It's better to think of all of these algorithms as being  $O(N + M \log M)$ , where M is the number of cells which pass through the priority-queue. M decreases dramatically between Barnes (2014a) and Wei (2018), as shown in Wei (2018, Figure 8). By the time Wei (2018), the O(N) term might even dominate the  $O(M \log M)$  term, meaning that Wei (2018) is a linear-time algorithm! Indeed, your Figure 5a suggests this. In any case, Figure 5c seems to overstate your results since it's not a proper performance comparison.

A: This point is very interesting. One major difference between our work and previous work is that we apply our algorithm on landscape evolution models, rather than as a filling processing step on real, acquired data. As such, we work on synthetic data where the number of local minima can be arbitrary important. One of our finding here is that there is a critical proportion of local minima where the original implementation (Barnes 2014) outperforms any subsequent work. An intuition of the reason of this effect can be given by changing the reviewer notation to N: the number of nodes outside of depressions and M: the number of nodes inside the depressions. We try to give a better analysis in the paper, but a very simplified intuition could be that Barnes (2014)'s complexity "looks like"  $O(M + N \log N)$ , while subsequent work "looks like"  $O(N + M \log M)$ . There are obviously some constraints on M and N that makes the analysis more complex, but the simplicity of Barnes algorithm makes its O(M) part extremely efficient, which is shown on fig 5.a. This raises two important questions: are our results globally better than more recent work, and in which cases our algorithm (and recent work), performs better than Barnes (2014); a study that seems to lack in Zhou or Wei's papers. This is the point of figure 5. b and c, that shows that in a typical fastscape model, our algorithm is in condition to perform better (in term of number of local minima), than Barnes'. The first question (whether our algorithm performs better than previous work, being Zhou 2017 or Wei 2018), is partially answered in fig 5.a. We choose to add more focus on this behavior analysis than a detailed performance comparison, because it gives the reader more hints on when to choose a method or another, and a performance evaluation, as noted by the referee, largely depends on a variety of external aspects such as optimization or adaptation of the algorithm to the computer architecture.

R: Added (Section 3.3 ; to justify the comparison with Barnes (2014a)): "Interestingly, the simplicity of this first version makes it the most efficient when the number of local minima is large (see Section 4).". Besides, we also added Wei (2018) in Fig 5c.

Q: p. 12 and 13: Given this description, I don't see how your benchmarks will be meaningful. Using the Wei (2018) algorithm in conjunction with the Barnes (2014b) algorithm for the benchmarks of Figure 5a makes a certain sense, but why not simply modify the Wei (2018) algorithm using the PF+e strategy of Barnes (2014a)? Wouldn't this comparison be more fair?

A: I double checked the code, we do not include the flat resolution algorithm on the benchmark. We add, however, a post process connectivity computation (receivers, donors and stack), which is, however, an important output of our algorithm. This is performed only for the purpose of comparing performance, because the connectivity computation on flat surfaces does not cancel local minima.

A2: We eventually found out, and we deeply apologize for that, that the above answer was wrong. The addition of the epsilon correction is important to compare algorithms with identical output, with state of the art approved methods. We also checked, and will mention it, that in practical cases, the algorithm from Wei (2018) without correction has performances equivalent to our results.

R: Added (Section 3.3): "Note that without combining it with a flat resolution algorithm, Wei (2018) algorithm shows an equivalent performance to our approach, provided that the depressions remain evenly distributed. In that case, the main difference between the two approaches is that ours provides data structures (flow paths, basin graph) that might be reused elsewhere."

Q: p. 13, Line 8: "we reuse the implementations available in the RichDEM" This seems reasonable, but it should be done with caveats. As you note on p. 2, this set of algorithms has "been optimized so that they can deliver acceptable performance when used with large datasets." It seems as though your algorithm uses significantly more space than the B, Z, and W algorithms you cite in your paper. The Barnes (2014a) algorithms work in-place using only the priority-queue and a boolean array as extra memory; that is, it requires about O(sqrt(N)) additional space. In comparison, your code has a Receivers array, Donors array (which takes O(8N) space), and Union-Find array, leading to a O(10N) space requirement. Trading space for time is a well-known optimization technique. This raises the question of how the B,Z,W algorithms would perform had they been optimized for small datasets. It also raises the question of how your algorithm would perform on larger datasets. The average size of the test dataset used by Wei (2018), for instance, was 2\*10^8 cells - two orders of magnitude larger than the 1500x1500 grid you use and three orders of magnitude larger than the Figure 5a benchmark. I don't think additional tests are necessary, but it should be clarified somewhere that the algorithms you're comparing against are being used for dataset sizes they weren't designed for.

A: We do not agree with this point. The donor & receiver data structures are still needed for many landscape evolution models using upwind numerical schemes), independently of the algorithm used, and the lack of optimization in this regard (donors array could be a N sized array) is more in favor of B, Z, W algorithms than ours. The union find and graph data structure are needed only for the depression graph, which size is generally much smaller than the grid size. Furthermore, we use grid size in the order of (or slightly larger than) what is generally used in landscape evolution modeling. We tried to benchmark larger data, up to 16k x 16k, which shows consistent results, but with more noise due to cache misses and other OS/material related issues that makes the results less readable.

**R**: Added (Section 3.3): "Note that because of the differences in the design/implementation used in RichDEM vs. our code, the benchmarks presented here should be seen as an illustration of the theoretical complexities of the algorithm variants rather than a strict comparison of their actual performances." And we eventually obtained clean benchmarks for 16k arrays that are now shown in Fig 5.

Q: p. 14, Lines 10-15: This is a really nice analysis. However, I'm not sure I agree with it entirely. This line: "Because those variants are more complex, k4 and k5 have higher values." Why is the cost of using a priority-queue higher for the Zhou and Wei algorithms? I would think that k2=k5. I agree that k4>k1. However, one of the reasons priority queues are expensive is because they have poor cache coherence. (Search online for "Latency numbers every computer programmer should know". See also, Luengo-Hendriks (2010)'s discussion of priority queue performance variation.) This means I expect k1

A: Yes, there is numbering issue,  $k_3 > k_0$  and  $k_4 > k_1$ . Cache coherency is an inherent problem of these depression filling algorithms, because of the random location of the depression on the topography, and which is hard to include in the discussion. Furthermore, the ratio of cache size to the grid size depends on the hardware and might change drastically in the next few years.

**R**: (Section 4) We fixed the numbering issue.

Q: Lines 10-15: Some analysis of the space used by the various algorithms would improve this discussion.

R: Added (end of Section 4): "For all the algorithms compared here, the memory consumption grows linearly with the DEM size. Barnes (2014a), Zhou (2016) and Wei (2018) Priority-Flood variants only use a supplementary queue to unload the priority queue, making it very memory-efficient. By contrast, our algorithm store more information like the structures used for representing flow paths (receivers, donors and stack), the basin graph and possibly some additional data structures like those needed by Mares (2002)'s algorithm. Some of this data might be required for further processing, *e.g.* the flow paths in landscape evolution modeling applications. Other data related to the basin graph increases the memory consumption, although in practice the number of local depressions – and thus the size of the graph – is small enough with respect to the size of the grid, resulting in only a small memory overhead compared to the Priority-Flood variants."

Q: Algorithm 6: It might be useful to note somewhere that the link is undirected such that Links.contains((n1,n2)) = = Links.contains((n2,n1)). The current implementation seems to rely on the ordering of the stack to ensure this is true, but in another context it may be useful to relax that ordering.

[revised manuscript text omitted]